# The Value and Optimal Sizes of Energy Storage Units in Solar-Assist Cogeneration Energy Hubs

**Xiaotao Chen [1], Yang Si [1,2], Chengkui Liu [3,4], Laijun Chen [1,\*], Xiaodai Xue [2], Yongqing Guo [1] and Shengwei Mei [1]**

1   Qinghai Key Lab of Efficient Utilization of Clean Energy (New Energy Photovoltaic Industry Research Center), Qinghai University, Xining 810016, China; chenxiaotao@qhu.edu.cn (X.C.); siyang@qhu.edu.cn (Y.S.); 2014990048@qhu.edu.cn (Y.G.); meishengwei@tsinghua.edu.cn (S.M.)
2   China State Key Laboratory of Power System and Generation Equipment, Department of Electrical Engineering, Tsinghua University, Beijing 100084, China; xuexiaodai@mail.tsinghua.edu.cn
3   Qinghai Building and Materials Research Co, Ltd., Xining 810008, Qinghai, China; helton8221@hotmail.com
4   The Key Lab of Plateau Building and Eco-community in Qinghai, Xining 810008, Qinghai, China
\*   Correspondence: chenlaijun@qhu.edu.cn; Tel.: +86-0971-511-4914

**Abstract:** Cogeneration is becoming increasingly popular in building and community energy systems with demands on electricity and heat, which is suitable for residential and industrial use in remote areas. This paper considers a stand-alone cogeneration energy hub. The electrical and thermal energies are produced by a combined heat and power (CHP) unit, photovoltaic panels, and a solar thermal collector. Since solar units generate no electricity and heat during the night, energy storage units which shift demands over time can promote the usage of solar energy and reduce the fuel cost of the CHP unit. This paper proposes a method to retrieve the optimal operation cost as an explicit function in the capacity parameters of electric and thermal energy storage units, reflecting the value of energy storage in the cogeneration energy hub. The capacity parameter set is divided into a collection of polyhedrons; on each polyhedron, the optimal value is an affine function in the capacity parameters. Furthermore, the optimal sizes of system components are discussed. The capacity of the CHP unit is determined from a linear program, ensuring supply adequacy; the capacities of solar generation and energy storage units are calculated based on the cost reduction and the budget. Case studies demonstrate the effectiveness of the proposed method.

**Keywords:** combined-heat-and-power generation; energy hub; energy storage unit; solar energy

## 1. Introduction

Two main energy demands in the daily life of mankind are electricity and heat. Compared with the separated production mode, combined heat and power (CHP) generation enjoys a higher overall efficiency [1] because the waste heat in electricity generation is utilized to produce heat. Distributed renewable energy resources, such as rooftop photovoltaic (PV) panels and solar thermal collectors, show various benefits in reducing the operation cost of CHP units and carbon dioxide emissions, and have been widely adopted in building energy systems [2]. However, solar produces no energy during the night, maintaining power and energy balance requires additional energy storage units. In addition, a thermal system has large inertia and heat demand is often flexible, so the heating system can act as a flexible load of the electric system. The integration of cogeneration plants, electric system, and heating system promotes the concept of the energy hub, which has been an active research direction since the pioneering work in [3,4].

### 1.1. Literature Review

An energy hub is an integrated facility with multiple input and output energy resources as well as energy conversion and storage units. Current research is mainly devoted to the operation and planning of energy hubs using different optimization methods.

From the operational perspective, based on the classical matrix description of energy hubs developed in [3,4], a general structure for energy hubs with renewable energy and power, heating, and cooling demands is put forward in [5]; the day-ahead operation problem is formulated as a mixed integer linear program considering demand response. The optimal network dispatch of a multi-energy system connected through energy hubs is studied in [6]; the gas transmission equation is approximate via multidimensional piecewise linear functions, so the network dispatch comes down to a mixed integer linear program, which can be processed by off-the-shelf solvers. The operation of a concentrating solar power energy hub connecting a power distribution system and a district heating system is studied in [7]; the linearized branch flow model and the hydraulic-thermal model are employed to describe the two networks. Energy hubs including wind, solar, hydro, and gas inputs are discussed in [8]; a matrix coupling model is proposed to describe the energy flow in the hub. Demand response with reducible and deferrable electric load, as well as flexible thermal load, is incorporated in energy hub scheduling problem in [9]; integrated demand response of power and heat is considered in the operation of networked energy hubs in power and natural gas systems [10]; a potential game is developed to capture the interactive behavior among energy hubs. A residential energy hub is a particular scenario for the multi-energy system without a network. Management of a residential energy hub comprising a CHP unit and an electric vehicle as battery storage is considered in [11], while in [12], the CHP unit is replaced with a concentrating solar power plant. The residential energy hub considered in [13] incorporates solar electricity and heat generation. A rule-based control method is reported in [14] for optimizing daily operational cost of an energy hub comprised of a CHP unit and battery/heat storage units.

To deal with uncertainty in energy hub operation, Parisio et al. in [15] develop a linear model for energy hub dispatch; uncertainty of demand is handled by a static robust optimization approach with relaxed equality constraints. A similar robust optimization method is also used in [16,17]. A stochastic programming technique is used in [18] to address the uncertain factors in energy hub operation, such as demand and market prices; it is found that the participation of heat energy in the market environment influences the optimal operation point. A risk-averse stochastic dynamic programming model for dispatching an energy hub is proposed in [19], in which electric and heat demands and market prices are modeled by stochastic processes. A chance-constrained stochastic optimization model is formulated in [20] for energy hub operation; the Cornish–Fisher expansion is adopted to transform chance constraints into deterministic ones. A probabilistic optimization model of renewable-powered residential energy hubs is suggested in [21]; the two-point estimate method is incorporated to mimic the uncertainty of rooftop solar generation. In short, the stochastic optimization model assumes an exact empirical distribution, which may be difficult to obtain in practice because the historical data are usually limited; robust optimization does not refer to the probability distribution and optimizes system performance in the worst-case scenario, but it is sometimes thought to be conservative because the worst case rarely happens in practice. To achieve a proper balance between reliability guarantee and conservativeness, distributionally robust optimization is introduced in the energy hub operation problem [22]; the stochastic output of solar output is modeled via a family of inexact probability distributions; the total operation cost in the worst-case distribution is optimized, and the problem finally boils down to a semidefinite program.

When there are multiple hubs, data privacy may be a concern, and distributed operation is desired. The interactive dispatch of multiple energy hubs is modeled as a potential game in [23], a distributed algorithm is developed to locate the Nash equilibrium based on the theory of potential game. In a standard Nash game, the strategy set of each player is decoupled. To incorporate coupled constraints which may impose an upper bound on the total import energy, a generalized Nash game model is

cast for the operation of networked energy hubs in [24], and a distributed algorithm is proposed for computing the generalized Nash equilibrium based on variational inequality theory and an improved Tikhonov regularization technique. A double-consensus distributed algorithm is developed in [25] to solve the optimal operation problem with multiple energy hubs; the method can cope with coupled variables in the objective function and constraint limits.

The operation of energy hubs in the market environment is attracting increasing attention recently. The bidding strategy of an energy hub in the day-ahead and real-time markets is studied in [26], and a mixed integer linear program is given to determine the optimal bidding strategy; uncertainty is modeled by a scenario tree. The strategic behavior of an energy hub in electricity and heat markets is discussed in [27] via a bilevel optimization model; the market-clearing process which captures the market power of the energy hub is explicitly modeled in the lower level. The similar problem for a cogeneration plan is considered in [28]; the alternating current power flow model is used for distribution market clearing, and the energy trading between the power distribution system and the district heating network is taken into account. The retail market led by an energy hub is considered in [29]; the energy hub purchases electricity and heat from upstream markets and resells energy to clients; the decision-making problem gives rise to a stochastic bilevel program, and is converted to a mixed-integer linear program. Such a framework is further studied in [30,31] by considering the competition among multiple energy hubs through a bilevel equilibrium program. Energy sharing among networked energy hubs is modeled in [32]; it is found that the sharing scheme can achieve almost the same efficiency as a centralized market.

From the planning perspective, the capacity expansion planning of energy hubs in multi-energy systems is firstly considered in [33]; the optimal sizes of gas furnaces and CHP units in the energy hubs, and capacities of generation units and transmission lines in the system, are determined from a mixed integer linear program. A mixed integer programming model is proposed in [34] to optimize the configuration and capacities of components based on graph theory. The method is applied to a multi-energy system in Beijing's new subsidiary administrative center [35]. An energy hub for a cement factory with electricity, heat and water demands is designed in [36]; exergy is taken into account to make full use of waste heat; economic and environmental impact is also analyzed. A reliability assessment method is proposed in [37], and a two-layer approach is developed for design of an energy hub; the upper layer aims to minimize investment cost, and the lower layer minimizes operation cost subject to reliability requirements in different scenarios. A bi-objective optimization method is presented in [38] to size the components in the energy hub and determine the maintenance cycles under different demand scenarios. In [39], a model for a CHP unit with improved accuracy is offered, and the CHP unit sizing problem integrating efficient operation under contingencies is modeled via a mixed integer linear program. A bi-level optimization model is presented in [40] for expansion planning of multi-energy system considering carbon emission constraints under a decentralized framework.

To better capture the uncertainty in the planning stage, a scenario-based stochastic programming model is proposed in [41] to design a wind-powered energy hub; reliability indices including loss-of-load expectation and expected energy not supplied are considered. The capacity planning of an energy hub with wind power, storage, and demand response is cast as a stochastic programming problem in [42]; uncertainty of wind generation is represented by 10 scenarios. Because the size of the optimization problem grows rapidly with an increase in the number of scenarios, a Benders decomposition method is proposed in [43] to reduce the computation burden. The stochastic optimization model is decomposed into a planning master problem and a collection of operation subproblems associated with scenarios; the impact of planning strategy on operation is reflected by the dual variables of subproblems. To relax the requirement on the exact probability distribution of uncertainty factors, the two-stage robust optimization method is applied in [44]; the uncertainty is restricted in an ellipsoid set; the first-stage determines the planning variables, including the capacities of energy hub components, and the second stage represents the operation under the given planning strategy; the life-cycle cost in the worst-case scenario is minimized. A similar optimization paradigm is also adopted in [45] to

co-optimize the planning of energy hubs in the integrated electricity and natural gas system with high penetration of volatile renewable generation; the uncertainty set is a polyhedral one, and in the problem solving, the lower-level problem is replaced by a Karush–Kuhn–Tucker (KKT) optimality condition. A distributionally robust optimization method for capacity planning of energy hub is proposed in [46]; the probability distributions of wind generation and demand are assumed to be close to respective empirical distributions in the sense of KL(Kullback-Leibler)-divergence, and the deterministic counterpart is derived based on conditional-value-at-risk and gives rise to a linear program.

Cogeneration system and energy hub have been a hot research topic for many years. Extensive research work can be found. We recommend comprehensive survey articles [47–50] for readers who wish to develop a holistic overview on this topic.

*1.2. Novelty and Contribution*

The contribution of this paper is twofold.

First, the value of energy storage in the energy hub is characterized by the optimal operation cost as a function in the capacity parameters. The property of this function is discussed. A sampling-based dual linear programming algorithm is proposed to retrieve the optimal value function. The sensitivity of the cost with respect to the storage capacity is explicitly reflected by the coefficients of the optimal value function. Such an outcome is informative and easy to visualize.

Second, the proper sizes of system components are determined from thorough optimization models. A linear programming model is proposed to size the CHP unit, ensuring supply adequacy even in the situation of continuous bad weather. Given an available budget, the optimal capacities of the PV panel, solar collector, and energy storage units can be calculated from a simple linear program based on the optimal value function.

## 2. System Configuration and Operation

The configuration of the solar-assist co-generation energy hub is shown in Figure 1. A gas-fired CHP unit is the main supply facility, which produces electricity and heat at the same time. PV panels and the solar thermal collector (STC) are equipped to harness solar energy. The excessive electricity and heat can be stored in the electric energy storage (EES) unit and thermal energy storage (TES) unit. The electric and thermal demands must be fully satisfied. Load shedding is not allowed. In the current operating practice of many distribution systems, reverse power flow is prohibited due to relaying protection-related considerations. So we do not consider selling excessive solar energy to the power grid. This problem is expected to be solved in modern active distribution systems.

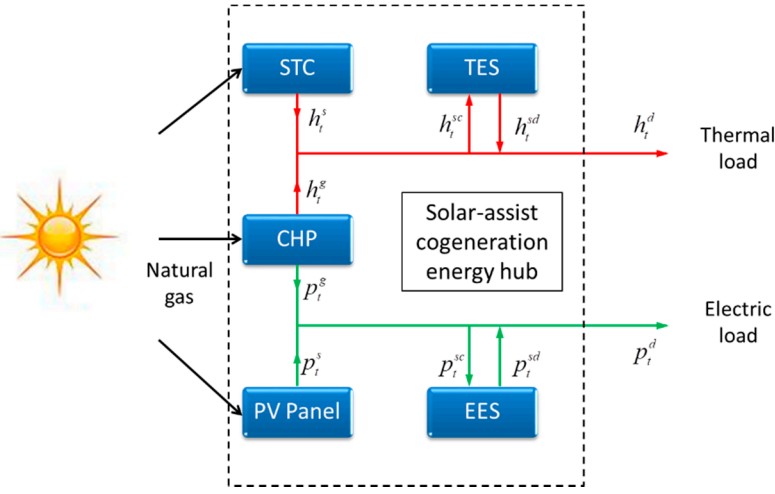

**Figure 1.** Configuration of the solar-assist co-generation energy hub.

Given the generation profiles of PV panel ($p_t^s$) and STC ($h_t^s$), as well as load curves $p_t^d$ and $h_t^d$, the daily operation problem of the energy hub can be cast as:

$$\min \sum_t C(p_t^g, h_t^g)\Delta t \tag{1.1}$$

$$\text{s.t. } p_t^g \geq 0, \ 0 \leq h_t^g \leq \alpha_h^g C_g, \ \frac{p_t^g}{\eta_p^g} + \frac{h_t^g}{\eta_h^g} \leq C_g, \ \forall t \tag{1.2}$$

$$p_t^g + p_t^s - p_t^{sc} + p_t^{sd} \geq p_t^d, \ \forall t \tag{1.3}$$

$$h_t^g + h_t^s - h_t^{sc} + h_t^{sd} \geq h_t^d, \ \forall t \tag{1.4}$$

$$\beta_l^e E_S \leq E_0 + \sum_{j=1}^{t}\left(\eta_{lc}^e p_j^{sc} - \frac{p_j^{sd}}{\eta_d^e}\right)\Delta t \leq \beta_m^e E_S, \ \forall t \tag{1.5}$$

$$\beta_l^h H_S \leq H_0 + \sum_{j=1}^{t}\left(\eta_c^h h_j^{sc} - \frac{h_j^{sd}}{\eta_d^h}\right)\Delta t \leq \beta_m^h H_S, \ \forall t \tag{1.6}$$

$$0 \leq p_j^{sc}, p_j^{sd} \leq \beta_{em}^{cd} E_S, \ 0 \leq h_j^{sc}, h_j^{sd} \leq \beta_{hm}^{cd} H_S, \ \forall t \tag{1.7}$$

where $p_t^g$ and $h_t^g$ are electric and thermal output of CHP unit, and the fuel cost of CHP unit is [51].

$$C(p, h) = \alpha_0 + \alpha_1 p + \alpha_2 h + \alpha_3 p^2 + \alpha_4 h^2 + \alpha_5 ph \tag{1.8}$$

Objective function (1.1) aims to minimize the daily fuel cost of the CHP unit. Because $C(p, h)$ is a convex function [51], it can be approximated via a piecewise linear function using the convex combination method in Appendix B of [52], and we can assume the objective function is linear without loss of generality. $C_g$ is the capacity of CHP unit, and $\alpha_h^g$ is a constant; $\eta_p^g / \eta_h^g$ is the efficiency of electricity/heat conversion; (1.2) defines the operating region of the CHP unit [46]. (1.3) and (1.4) are supply adequacy constraints, where power flow variables are clarified in Figure 1, and inequality means that excessive renewable production can be curtailed. (1.5) and (1.6) describe the feasible ranges of the state-of-charges (SoCs) of EES and TES units, where $E_S / H_S$ and $E_0 / H_0$ are the capacity and initial SoC of the EES/TES unit; $\eta_c^e / \eta_d^e$ and $\eta_c^h / \eta_d^h$ are charging/dispatching efficiency of the EES/TES unit; $\beta_l^e / \beta_m^e$ and $\beta_l^h / \beta_m^h$ are constant coefficients indicating the minimum and maximum storage level of EES and TES units. (1.7) restricts the maximum charging and discharging rates of the EES/TES unit, where $\beta_{em}^{cd}$ and $\beta_{hm}^{cd}$ are constant coefficients; their multiplicative inverse indicates how long it takes to charge the storage units from empty to full. In problem (1.1)–(1.7), decision variables include power flow variables of the CHP unit and energy storage units. This simulates daily operation with 24 periods. The first period is 0:00–1:00 a.m., shortly after the evening peak, so $E_0 / H_0$ is set to the minimum SoC. We can also relax the initial condition and let $E_0 / H_0$ be a decision variable. Nevertheless, if the initial condition is set too high, more energy should be purchased during the evening peak, causing a rise in the operation cost.

Define $\theta = \begin{bmatrix} E_S & H_S \end{bmatrix}^T$ and $x = \begin{bmatrix} p_t^g, h_t^g, p_t^{sc}, p_t^{sd}, h_t^{sc}, h_t^{sd} \end{bmatrix}$, $\forall t$, problem (1.1)–(1.6) can be written in a compact matrix form as:

$$\begin{aligned} \min \ & c^T x \\ \text{s.t. } & Ax \leq b + B\theta \end{aligned} \tag{2}$$

where $A$, $B$, $b$ and $c$ are constant matrices and vectors corresponding to the coefficients in problem (1.1)–(1.7). The capacity parameter set is:

$$\Theta = \left\{ \theta \ \middle| \ \begin{array}{l} 0 \leq E_S \leq E_M \\ 0 \leq H_S \leq H_M \end{array} \right\} \tag{3}$$

where $E_M$ and $H_M$ are maximum capacities of EES and TES, respectively.

## 3. Quantifying the Value of Energy Storage

The compact daily operation problem is given in (2). On each day, the problem data is different. Suppose we have the data in $s = 1, \cdots, S$ typical days, which refer to the solar generation profiles and load curves that affect vector b in the constraints, while *A*, *B* and *c* remain unchanged. Therefore, we use a superscript s to distinguish the problem associated with day s:

$$
\begin{aligned}
v^s(\theta) = \min\ & c^{\mathrm{T}} x^s \\
\text{s.t. } & A x^s \leq b^s + B\theta
\end{aligned}
\tag{4}
$$

The optimal value is a function of θ. If the probability of scenario s is $\rho^s$, the average cost is:

$$
v(\theta) = \sum_{s=1}^{S} \rho^s v^s(\theta)
\tag{5}
$$

If the charging and dispatching power is fixed at 0, the feasible set is the same as the one with $\theta = 0$, hence for any $\theta \geq 0$, the feasible region is larger, and $v(0) \geq v(\theta)$, $\forall \theta \geq 0$ holds. The value of energy storage is defined as the cost reduction:

$$
V(\theta) = v(0) - v(\theta)
\tag{6}
$$

In the rest of this section, we develop a linear programming based method to retrieve an approximate expression of $v(\theta)$ in closed form. In what follows, the average cost problem,

$$
\begin{aligned}
v(\theta) = \min\ & \sum_{s=1}^{S} c^{\mathrm{T}} x^s \\
\text{s.t. } & A x^s \leq b^s + B\theta, \ \forall s
\end{aligned}
\tag{7}
$$

is denoted by:

$$
\begin{aligned}
v(\theta) = \min\ & \sum_{s=1}^{S} \bar{c}^{\mathrm{T}} \bar{x} \\
\text{s.t. } & \overline{A}\bar{x} \leq \bar{b} + \overline{B}\theta
\end{aligned}
\tag{8}
$$

where

$$
\bar{c} = \begin{bmatrix} c \\ \vdots \\ c \end{bmatrix}, \bar{x} = \begin{bmatrix} x^1 \\ \vdots \\ x^s \end{bmatrix}, \overline{A} = \begin{bmatrix} A & & \\ & \ddots & \\ & & A \end{bmatrix}, \bar{b} = \begin{bmatrix} b^1 \\ \vdots \\ b^s \end{bmatrix}, \overline{B} = \begin{bmatrix} B \\ \vdots \\ B \end{bmatrix}
\tag{9}
$$

The dual problem of LP (8) is:

$$
\begin{aligned}
v(\theta) = \max_{\mu \in U}\ & \mu^{\mathrm{T}}(\bar{b} + \overline{B}\theta) \\
U = \Big\{ \mu \Big| & \overline{A}^{\mathrm{T}} \mu = \bar{c}, \mu \leq 0 \Big\}
\end{aligned}
\tag{10}
$$

where $\mu$ is the dual variable, and $U$ is the feasible set of dual variables. Because strong duality always holds for linear programs, the optimum of problem (10) is $v(\theta)$ for any given θ. For a fixed $\mu \in U$, $\mu^T(\bar{b} + \overline{B}\theta)$ is an affine function in θ. LP (10) states that $v(\theta)$ is the pointwise maximum of affine functions $\mu^T(\bar{b} + \overline{B}\theta)$ when $\mu$ takes all possible values in $U$, so $v(\theta)$ is convex in θ, because point-wise maximum preserves convexity.

Actually, the enumeration over $U$ with an infinite number of elements can be replaced with the enumeration over the vertex set vert($U$) whose number of elements is finite, i.e.,:

$$
v(\theta) = \max_i \big\{ \mu_i^{\mathrm{T}} \bar{b} + \mu_i^{\mathrm{T}} \overline{B}\theta \big\}, \ \forall \mu_i \in \text{vert}(U)
\tag{11}
$$

Nevertheless, the majority of the vertices will produce redundant pieces, which never reaches the maximum. Next, we define the minimum representation of $v(\theta)$.

**Definition 1.** *(Non-redundant vertex) a vertex $\mu_i \in \text{vert}(U)$ is called non-redundant if the affine function $\mu_i^T(\bar{b} + \bar{B}\theta)$ reaches the maximum for some $\theta \in \Theta$.*

According to Definition 1, vert($U$) can be divided into two subsets; one consists of non-redundant vertices and the other redundant ones. The former is denoted by $\bar{V}(U)$.

**Definition 2.** *The minimum representation of $v(\theta)$ is:*

$$v(\theta) = \max_i \{m_i + n_i^T\theta\}, \ \forall \mu_i \in \bar{V}(U) \tag{12}$$

*where $m_i = \mu_i^T\bar{b}$ and $n_i = \mu_i^T\bar{B}$ are constant coefficients. Another representation of the piecewise linear (PWL) function in (12) is as follows:*

$$v(\theta) = \begin{cases} m_1 + n_1^T\theta, & \theta \in \text{CR}_1 \\ \vdots & \vdots \\ m_k + n_k^T\theta, & \theta \in \text{CR}_k \end{cases} \tag{13}$$

*where $\text{CR}_1, \cdots, \text{CR}_k$ are called critical regions, and $\Theta = \cup_{i=1}^k \text{CR}_i$. Critical regions are the set of parameters in which respective piece reaches the maximum, therefore*

$$\text{CR}_i = \left\{\theta \middle| m_i + n_i^T\theta \geq m_{[-i]} + n_{[-i]}^T\theta\right\}, [-i] = \{1, \cdots, i-1, i+1, \cdots, k\} \tag{14}$$

If $v(\theta)$ is not the minimum representation and contains redundant vertices, then some critical region will be empty because the inequality set does not have a solution. To test if a vertex is redundant, we solve the following linear program

$$\begin{aligned} \min_{\theta, s} \ & s \\ \text{s.t. } & m_i + n_i^T\theta + s \geq m_{[-i]} + n_{[-i]}^T\theta \\ & s \geq 0 \end{aligned} \tag{15}$$

If the optimal value is equal to zero, then the vertex $\mu_i$ is non-redundant. Optimal value function (13) provides explicit sensitivity information. The marginal cost reduction brought by the per unit increase of storage capacity is:

$$\frac{\partial v(\theta)}{\partial \theta} = \begin{cases} n_1 & \theta \in \text{CR}_1 \\ \vdots & \vdots \\ n_k & \theta \in \text{CR}_k \end{cases} \tag{16}$$

However, the vertex set vert $(U)$ not known in advance. We propose a sampling-based algorithm to generate an approximation of $v(\theta)$. The flow chart is summarized as follows:

---

**Algorithm 1 Constructing the Optimal Value Function**

---

1: **Step 1**: Create uniformly distributed grid points $\theta_1, \cdots, \theta_l$ in the parameter set $\Theta$.

2: **Step 2**: Solve linear program (10) for each $\theta_i$; the corresponding optimal solution is $\mu_i$.

3: **Step 3**: Remove duplicated dual variables, and calculate $m_i = \mu_i^T\bar{b}$ and $n_i = \mu_i^T\bar{B}$; construct
$v(\theta) = \max_{i=\{1,\cdots,k\}}\{m_i + n_i^T\theta\}$

4: **Step 4**: Redundancy screen. For each linear function $m_i + n_i^T\theta$ in the optimal value function, solve.LP (15).
If the optimal value is strictly positive, then remove this linear function from $v(\theta)$.

5: **Step 5**: Construct critical regions according to (14).

---

According to our experiments, using $10 \times 10$ grid points as the initial samples, Algorithm 1 can produce high quality approximation for $v(\theta)$. In problem (10), $\theta \in \mathbb{R}^2$, and x is a high-dimensional vector. The average time for solving problem (10) is 0.1 s If we use $10 \times 10$ grid points, the computation time of Algorithm 1 is less than half a minute.

The proposed method is developed based on the abstract formulation (2) and makes no specific assumptions on the objective function and operating constraints, except for linearity. On this account, Algorithm 1 is also valid if the linear objective function considers carbon emission or the energy loss/solar power curtailment, or the bidirectional electricity arbitrage is modeled. Future expansions of system facilities can be considered as well, by simply modifying the baseline value of the parameter vector $\theta$.

## 4. Sizing System Components

Previous discussions are based on fixed system configuration. One possible application of the outcome is to determine the proper capacities of components in the energy hub, which is addressed in this section.

### 4.1. Sizing the Combined Heat and Power (CHP) Unit

Since solar units produce no energy in bad weather or during the night. To guarantee supply adequacy even in the situation of continuous bad weather, we assume the CHP unit must be able to supply the electricity and heat demands without solar input. However, the electric power and thermal outputs are not separable, the capacity of CHP unit cannot be determined straightforward. In this regard, we solve the following linear program:

$$\min C_g$$
$$\text{s.t. } h_t^d \le \alpha_h^g C_g, \ \frac{p_t^d}{\eta_p^g} + \frac{h_t^d}{\eta_h^g} \le C_g, \ \forall t \tag{17}$$

where the objective function is to find the minimum capacity subject to supply adequacy constraint. The electrical and thermal capacities of the CHP are $\eta_p^g C_g$ and $\alpha_h^g C_g$, respectively. Such a methodology makes problem (2) feasible for all $\theta \in \Theta$, and thus $v(\theta)$ is always finite.

### 4.2. Sizing Energy Storage Units with Fixed Solar Capacities

The rooftop PV panels, STC, and energy storage units are used to reduce the operation cost of the CHP unit, if weather permits. Now we consider the situation in which the capacities of PV panels and STC have been given. The capacities of EES and TES units are to be determined. Let $\Gamma$ be the available budget, and $c_E$ and $c_H$ the unit capacity costs of EES and TES, and then we solve:

$$\min_{\sigma, \theta} \sigma$$
$$\text{s.t. } \sigma \ge m_i + n_i^T \theta, \ \forall i \tag{18}$$
$$c_E \theta_1 + c_H \theta_2 \le \Gamma, \ \theta \in \Theta$$

The optimal solution $\theta^*$ gives the capacities of EES and TES that lead to the minimum cost subject to the budget of investment. Recall the cost reduction function in (6), the cost recovery time is:

$$T = \frac{\Gamma}{V(\theta^*)} \text{ days} \tag{19}$$

The proper investment intensity can be determined by comparing the cost recovery time under different budgets.

### 4.3. Sizing Solar and Energy Storage Units

The investment on PV panels and STC can be considered as well by slightly revamping the parametric programming model. First, (1.3) and (1.4) is modified as:

$$p_t^g + \alpha_E p_t^s - p_t^{sc} + p_t^{sd} \geq p_t^d, \ \forall t \tag{20}$$

$$h_t^g + \alpha_H h_t^s - h_t^{sc} + h_t^{sd} \geq h_t^d, \ \forall t \tag{21}$$

where $p_t^s$ and $h_t^s$ are the output of per unit capacity PV panel and STC; $\alpha_E$ and $\alpha_H$ are their respective capacity. Define capacity parameter $\theta = [\alpha_E \ \alpha_H \ E_S \ H_S]^T$ and parameter set as:

$$\Theta = \left\{ \theta \ \middle| \ \begin{matrix} 0 \leq \alpha_E \leq \alpha_{EM}, \ 0 \leq \alpha_H \leq \alpha_{HM} \\ 0 \leq E_S \leq E_M, \ 0 \leq H_S \leq H_M \end{matrix} \right\} \tag{22}$$

Problem in (1) with (20) and (21) can be written in a compact matrix form as (2). Algorithm 1 can be used to retrieve the optimal value function $v(\theta)$.

With the optimal value function $v(\theta)$, the optimal sizes of system components can be calculated from a model similar to (18)

$$\begin{aligned} \min_{\sigma, \theta} \quad & \sigma \\ \text{s.t.} \quad & \sigma \geq m_i + n_i^T \theta, \ \forall i, \theta \in \Theta \\ & c_{PV} \theta_1 + c_{STC} \theta_2 + c_{EES} \theta_3 + c_{TES} \theta_4 \leq \Gamma \end{aligned} \tag{23}$$

where $c_{PV}$ and $c_{CS}$ represent the unit capacity costs of PV panel and STC, respectively. The cost recovery time is the same as (19).

## 5. Case Studies

The proposed method is tested based on real data collected from an industrial park near the city of Xining, Qinghai Province, China. Electricity and heat demands in four typical days in spring, summer, autumn and winter are used. The maximum electricity/heat demand is 240 MW/200 MW, both happening in winter. The solar radiation data in 2019 is available at [53]. Parameters of system components are provided in Table 1. Using the model in Section 4.1, the CHP unit must have a rated electrical capacity of 273 MW and thermal capacity of 450 MW. As the electrical and thermal outputs of CHP unit are not independent, the capacity is larger than the respective maximum demands. The cost function is $C(p, h) = 0.09p^2 + 280p + 0.05h^2 + 70h + 0.03ph + 12000$ CNY.

**Table 1.** Parameters of System components.

| EES Unit | | TES Unit | | CHP Unit | |
|---|---|---|---|---|---|
| **Parameter** | **Value** | **Parameter** | **Value** | **Parameter** | **Value** |
| $\eta_c^e / \eta_d^e$ | 0.95/0.95 | $\eta_c^h / \eta_d^h$ | 0.98/0.98 | $\eta_p^g$ | 0.30 |
| $\beta_l^e / \beta_m^e$ | 0.20/0.95 | $\beta_l^h / \beta_m^h$ | 0.10/1.00 | $\eta_h^g$ | 0.90 |
| $\beta_{em}^{cd}$ | 0.20 | $\beta_{hm}^{cd}$ | 0.20 | $\alpha_h^g$ | 0.50 |

Next, we investigate the value of EES and TES. If the demand is served by the CHP, the storage is barely dispatched because of the inevitable energy losses. Only when solar energy output exceeds the demand could the value of energy storage be reflected. In this bunch of tests, we first examine the accuracy of the proposed method in Algorithm 1. The capacity of PV panel and STC is fixed at 300 MW. We use different number of initial samples. The surfaces and critical regions of the obtained optimal value function (OVF) are portrayed in Figure 2.

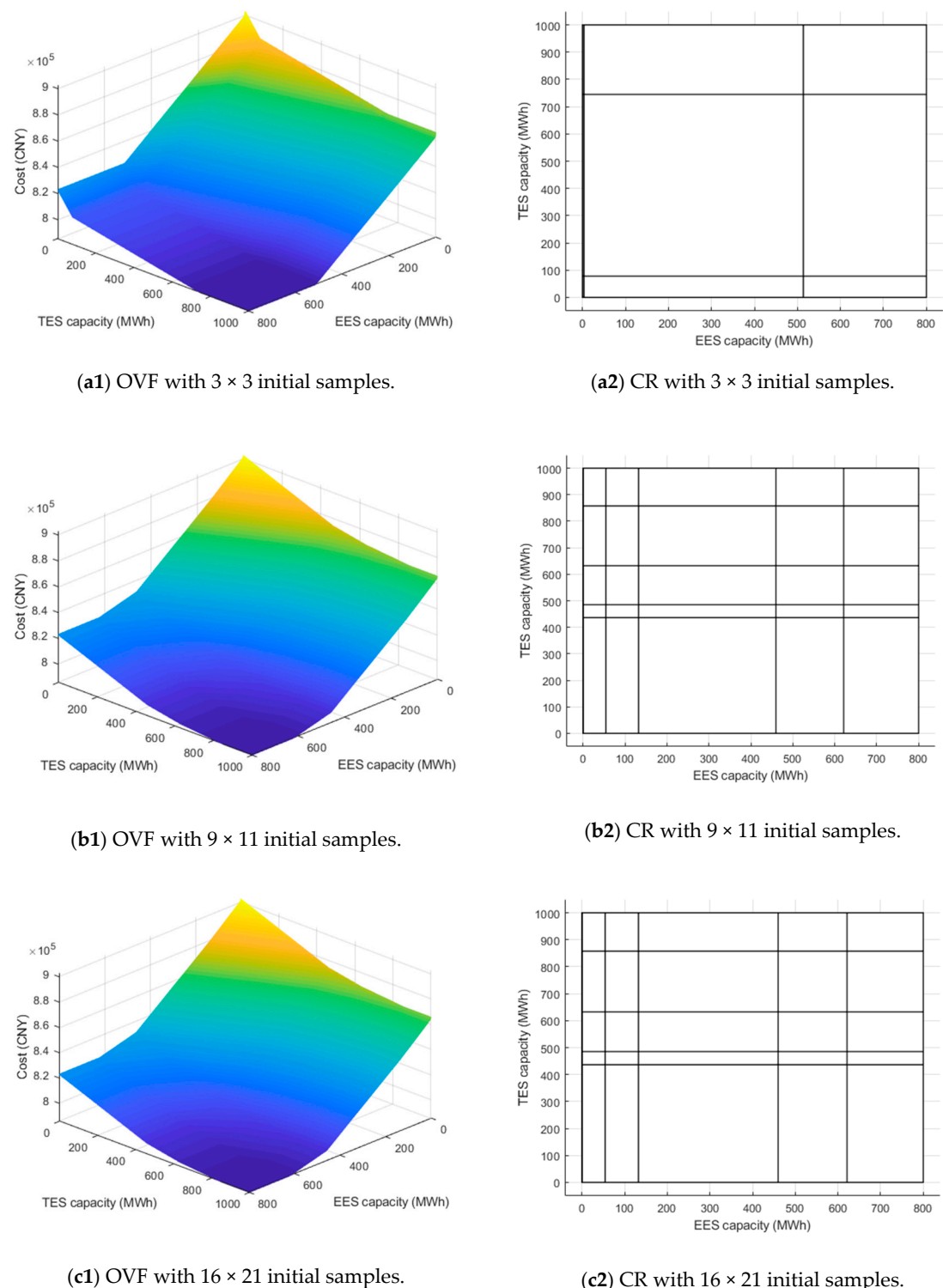

(**a1**) OVF with 3 × 3 initial samples. (**a2**) CR with 3 × 3 initial samples.

(**b1**) OVF with 9 × 11 initial samples. (**b2**) CR with 9 × 11 initial samples.

(**c1**) OVF with 16 × 21 initial samples. (**c2**) CR with 16 × 21 initial samples.

**Figure 2.** OVFs and critical regions (CRs) with different initial samples.

With 16 × 21 initial samples, the OVF retrieved from Algorithm 1 is exact. For verification, we solve problem (2) with 101 × 101 sampled values of θ, and compare the optimum with the optimal value offered by the OVF in Figure 2c, the maximum relative error is below $10^{-4}$. Then, we reduce the number of initial samples to 9 × 11; the same result is obtained, demonstrating that the proposed method does not rely on a set of very dense sampling points, and is robust once sufficient samples are

provided. However, if the number of samples is insufficient, say $3 \times 3$, the result is shown in Figure 2a: the approximate OVF underestimates the true one, and fewer critical regions are found.

From Figure 2, the value of energy storage is clear. Without energy storage, the operation cost is as high as 902,698 CNY. With the increase of the sizes of the EES unit and TES unit, the cost gradually decreases. When $E_S = 621.3$ MWh and $H_S = 857.7$ MWh, the cost reduces to the minimum of 785,671 CNY. Further enlarging storage capacity brings no additional benefit, as there is no more energy to be stored. It is also observed that the marginal cost reduction diminishes with the increase of storage capacity, which follows from the convexity of the OVF.

The impact of solar generation is investigated. The capacities of PV panel and STC are changed from 100 MW to 500 MW; the results are shown in Figure 3 and Table 2. When $C_{PV} = C_{STC} = 100$ MW, the electric load is always larger than solar power output, so EES is not dispatched. The cost decreases with a larger TES capacity, and reaches the minimum when $H_S = 105$ MWh; further increasing $H_S$ has no impact on the cost, because there is no surplus energy to be stored. The maximum cost reduction is only 0.2%. When $C_{PV} = C_{STC} = 200$ MW, the minimum cost is achieved at $E_S = 40$ MWh and $H_S = 570$ MWh. The cost reduction is mainly attributed to the TES unit. Nevertheless, the situation changes with the continuous growth of solar capacity. When $C_{PV} = C_{STC} = 400$ MW, an EES with 1292 MWh capacity and a TES with 949 MWh capacity are need to take full use of solar energy, and the 26.8% cost reduction is mainly attributed to the EES unit, which can be observed from Figure 3c.

Finally, we consider sizing the components in the energy hub. The unit capacity costs of PV panel, STC, EES and TES are $c_{PV} = 6$ million CNY/MW, $c_{STC} = 1$ million CNY/MW, $c_{EES} = 1.25$ million CNY/MWh, and $c_{TES} = 0.2$ million CNY/MWh. We change the available budget from 0 to 5 billion CNY. The optimal sizing strategy and optimal daily operation cost are retrieved from model (23). The results under different budgets are provided in Figure 4 and Table 3. When the budget is rare, all investment will be spent on PV panels and STC; storage is needed only if electricity or heat produced by solar energy is larger than the demand. When the budget is greater than 1.5 billion CNY, the marginal cost reduction brought by per unit investment decreases, because EES and TES should be invested in accompanying PV panels and STC. The efficiency of investment is also examined. For a fixed budget $\Gamma$, let the corresponding optimal sizing strategy be $\theta^*(\Gamma)$, then the cost recovery time can be calculated from $\Gamma/365(v(0) - v(\theta^*(\Gamma)))$ (in years). When the budget $\Gamma \leq 1.3$ billion CNY, the cost recovery time is 8.8 years, and grows at a constant rate with the increase of budget $\Gamma$.

Suppose the lifespan of the facility is 10 years, the net profit under an available budget $\Gamma$ and the corresponding optimal sizing strategy $\theta^*(\Gamma)$ is $365(v(0) - v(\theta^*(\Gamma))) - \Gamma$. The results are given in Figure 5, from which we can see that from an optimal net profit perspective, the optimal investment seems to be $\Gamma = 1.3$ billion CNY. If the lifespan of the facility can reach 20 years, then the optimal investment is $\Gamma = 2.0$ billion CNY. Results in Table 4 show that most of the investment on storage will be spent on the TES unit, because of its low cost, and hence the saved capacity of the CHP unit can be used to produce electricity.

In the future, the costs of battery and solar panel are expected to drop continuously. Investment on renewable energy and storage will become more rewarding. We test the impact of $c_{PV}$ and $c_{EES}$ on the sizing strategy and net profit. The lifespan of the facility is assumed to be 10 years, and the budget is fixed at 1.5 billion CNY. The optimal sizing strategy and optimal daily operation cost are retrieved from (23), and the results are summarized in Table 4. At the beginning, with the decrease of $c_{PV}$ and $c_{EES}$, the saved cost will be spent on solar panel, STC, and especially TES, which is verified by the rapidly increased capacity. With the help of EES, the CHP can have more capacity in producing electricity, so EES is not invested in. However, with the continuous decline of $c_{PV}$ and $c_{EES}$, EES becomes more favorable, as its cost reduction potential shown in Figure 3c can be fully unleashed; at the same time, the investment in STC and TES reduces.

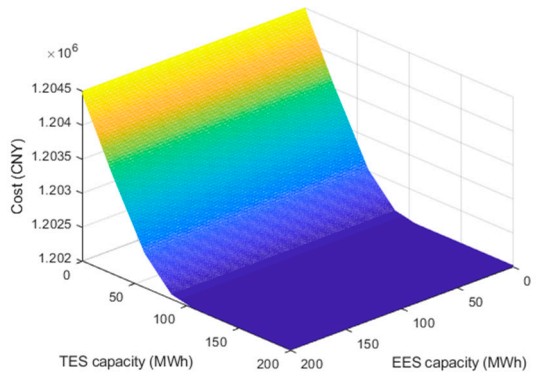

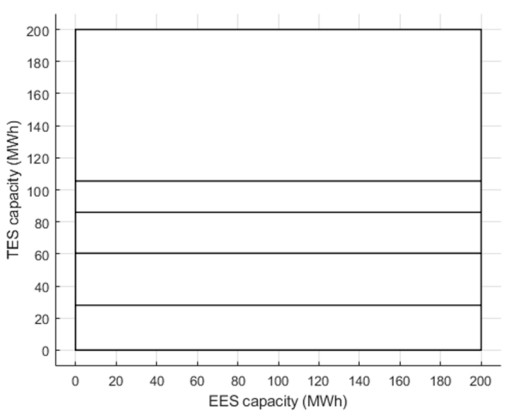

(**a1**) OVF with $E_S = 100$ MWh $H_S = 100$ MWh.

(**a2**) CR with $E_S = 100$ MWh $H_S = 100$ MWh.

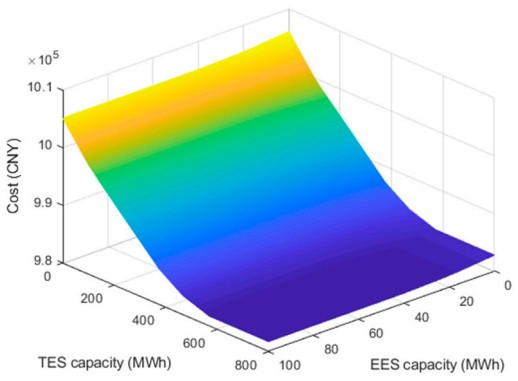

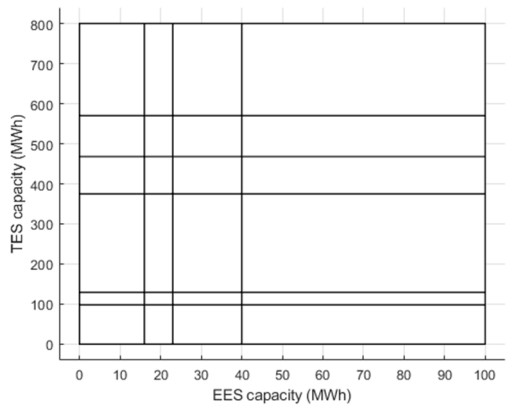

(**b1**) OVF with $E_S = 200$ MWh $H_S = 200$ MWh.

(**b2**) CR with $E_S = 200$ MWh $H_S = 200$ MWh.

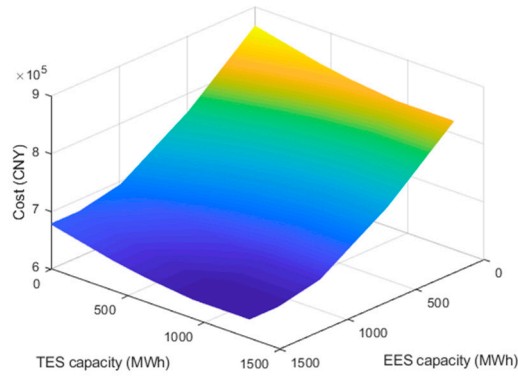

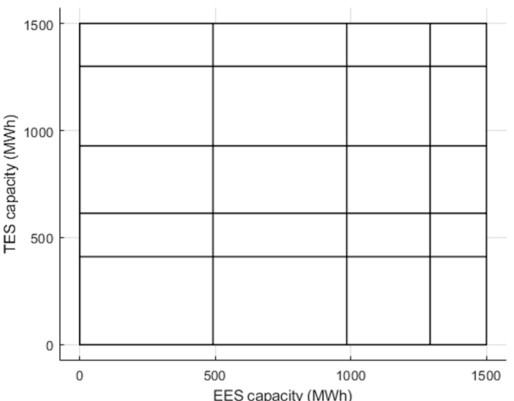

(**c1**) OVF with $E_S = 400$ MWh $H_S = 400$ MWh.

(**c2**) CR with $E_S = 400$ MWh $H_S = 400$ MWh.

**Figure 3.** OVF and critical regions with different solar capacities.

**Table 2.** Optimal storage sizes and economic performances with different solar capacities.

| PV Panel (MWh) | STC (MWh) | $V(0)$ CNY | $V_{min}$ CNY | $E_S$ (MWh) | $H_S$ (MWh) | Reduction % |
|---|---|---|---|---|---|---|
| 100 | 100 | 1,204,493 | 1,202,018 | 0 | 105 | 0.2 |
| 200 | 200 | 1,006,167 | 981,740 | 40 | 570 | 2.4 |
| 300 | 300 | 902,698 | 785,671 | 621 | 858 | 13.0 |
| 400 | 400 | 865,776 | 634,128 | 1292 | 949 | 26.8 |
| 500 | 500 | 846,545 | 588,265 | 1370 | 1279 | 30.5 |

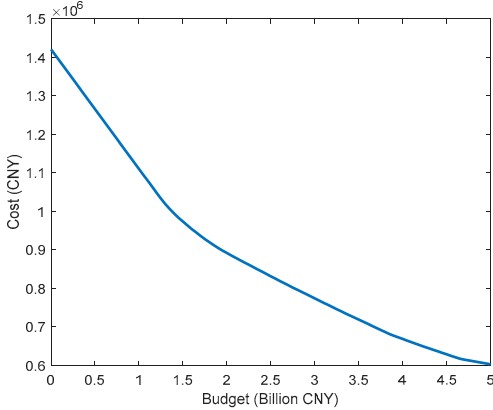

(**a**) Minimum cost as a function of budget.

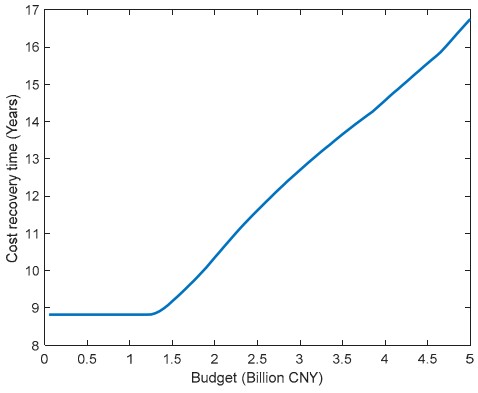

(**b**) Cost recovery time.

**Figure 4.** Relationship between budget and cost.

**Table 3.** Optimal sizing strategies with different budgets.

| Budget ($10^9$CNY) | PV Panel (MW) | STC (MW) | EES (MWh) | TES (MWh) |
|---|---|---|---|---|
| 0.5 | 70 | 80 | 0 | 0 |
| 1.0 | 153.3 | 80 | 0 | 0 |
| 1.5 | 231.7 | 106.8 | 0 | 15.95 |
| 2.0 | 270.3 | 241.1 | 20.3 | 559.8 |
| 2.5 | 283.2 | 259.7 | 328.7 | 650.7 |
| 3.0 | 315.4 | 259.7 | 574.3 | 650.7 |

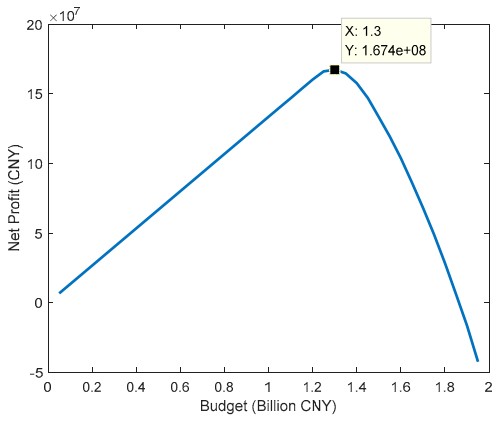

(**a**) In the period of 10 years.

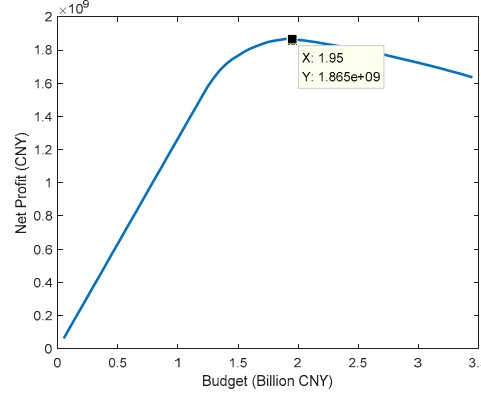

(**b**) In the period of 20 years.

**Figure 5.** Net profit in a certain period.

**Table 4.** Optimal sizing strategies with different costs of solar panel and EES.

| $c_{PV}$ CNY/MW | $c_{EES}$ CNY/MWh | PV Panel (MW) | STC (MW) | EES (MWh) | TES (MWh) | Payback Time (days) | Net Profit CNY |
|---|---|---|---|---|---|---|---|
| $5.6 \times 10^6$ | $1.2 \times 10^6$ | 240.6 | 133.9 | 0 | 92.6 | 3207 | $2.07 \times 10^8$ |
| $5.2 \times 10^6$ | $1.1 \times 10^6$ | 250.9 | 156.5 | 0 | 194.7 | 3080 | $2.78 \times 10^8$ |
| $4.8 \times 10^6$ | $1.0 \times 10^6$ | 264.9 | 174.9 | 0 | 267.5 | 3033 | $3.05 \times 10^8$ |
| $4.4 \times 10^6$ | $0.9 \times 10^6$ | 271.1 | 205.8 | 23.4 | 401.1 | 3030 | $3.07 \times 10^8$ |
| $4.0 \times 10^6$ | $0.8 \times 10^6$ | 280.6 | 133.9 | 275.4 | 117.1 | 3028 | $3.08 \times 10^8$ |

## 6. Conclusions

This paper studies a solar-assist cogeneration energy hub and the role of electrical and thermal energy storage in reducing the fuel cost. A linear programming-based algorithm is proposed to generate an approximate optimal value function in storage parameters, reflecting the value of energy storage in the energy hub; the algorithm can also tackle other parameters such as the capacities of solar panels and the solar thermal collector. A direct application of the above outcome is to determine the size of each component. Some phenomena are observed from case studies: solar panels and the solar thermal collector are firstly set up in the hub, when there is excessive energy generated by the solar units, energy storage is invested in. The optimal value functions suggest that the marginal benefit brought by energy storage decreases. If the capacity of energy storage is large enough, the electrical storage brings more benefit than the thermal storage. However, from an optimal net profit perspective, the optimal choice is to invest in cheap thermal energy storage if the budget is limited, so as to save the capacity of the CHP unit to produce electricity. Otherwise, if plenty of funds are available, investing in electric storage is more profitable.

**Author Contributions:** Conceptualization, X.C. and S.M.; Data curation, Y.S.; Formal analysis, C.L.; Methodology, L.C.; Supervision, S.M.; Validation, X.X.; Visualization, Y.G.; Writing—original draft, X.C.; Writing—review and editing, X.C. All authors have read and agreed to the published version of the manuscript.

**Funding:** This work has been supported by special fund project of The Key Lab of Plateau Building and Eco-community in Qinghai, China (KLKF-2019-004), Young Research Fund Project of Qinghai University (2019-QGY-1), and The Scientific and Technological Project of Qinghai Province (2018-ZJ-774).

**Conflicts of Interest:** The authors declare no conflict of interest.

## Nomenclature

The majority of symbols and notations used throughout the paper are defined below for quick reference. Others are clarified with their appearance in case of need.

| | |
|---|---|
| $\alpha_h^g$ | Constant coefficient related to the maximum thermal output of CHP unit. |
| $\alpha_0 \sim \alpha_5$ | Constant coefficients in the cost function of CHP unit. |
| $\beta_l^e / \beta_m^e$ | Constant coefficient indicating the minimum and maximum electric storage level. |
| $\beta_l^h / \beta_m^h$ | Constant coefficient indicating the minimum and maximum thermal storage level. |
| $\beta_{em}^{cd} / \beta_{hm}^{cd}$ | Energy to power ratio of electric/thermal energy storage, constant coefficient |
| $C_g$ | Capacity of CHP unit, parameter. |
| $E_S / H_S$ | Capacity of the electric/thermal storage unit, parameter. |
| $E_0 / H_0$ | Initial state-of-charge of the electric/thermal storage unit, depending on $E_S / H_S$. |
| $h_t^d$ | Heat load curve, problem data. |
| $h_t^g$ | Thermal output of CHP unit, decision variable. |
| $h_t^s$ | Generation profile of solar thermal collector, problem data. |
| $h_t^{sc} / h_t^{sd}$ | Charging/discharging power of thermal energy storage in period t. |
| $p_t^d$ | Electric load curve, problem data |
| $p_t^g$ | Electric output of CHP unit, decision variable. |
| $p_t^s$ | Generation profile of solar panel, problem data. |
| $p_t^{sc} / p_t^{sd}$ | Charging/discharging power of electric energy storage in period t. |

| $\eta_c^e / \eta_d^e$ | Charging/dispatching efficiency of electric energy storage, constant coefficient. |
| $\eta_c^h / \eta_d^h$ | Charging/dispatching efficiency of the thermal energy storage, constant coefficient. |
| $\eta_p^g / \eta_h^g$ | Efficiency of electricity/heat conversion of CHP unit, constant coefficient. |
| $\Delta t$ | Duration of period t. |

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
