# Peer review of "The Value and Optimal Sizes of Energy Storage Units in Solar-Assist Cogeneration Energy Hubs"

_applsci, doi:10.3390/app10144994_

Round 1
Reviewer 1 Report
The explanation of the research presented within this paper is clear. The results are coherent with previous knowledge. Nevertheless, I consider that several issues need additional information, and some justifications should be added to improve the quality of the manuscript.
- It should be very useful for understand correctly this research to include a nomenclature explaining all the variables, subscripts and superscripts.
- Your objective, aim to minimize the daily fuel cost is obviously correct and probably the most suitable, but your research should be extended considering for instance the energy efficiency or the carbon footprint as different objectives. These few changes in the algorithm would obtain totally different results. For example, the PV and STC operation should be considered equals? (from an energy efficiency perspective)?
- Line 340-341: ‘In the near future, investment on renewable energy and storage will become more rewarding with the continuous drop of their unit capacity costs‘. You recognise the no-negligible effect of these probable scenario changes. In my opinion your research should take into account different future scenarios, in the cost of natural gas, as well as the accumulation, TES and EES. For example, the possibility of increase the accumulation of energy (depend on the evolution of costs) in this period should be considered, as well as modifications to the current installation, which could have an evident impact on presented results.
- You should justify why it was not considered the selling or redistributing of the excess of energy produced, if it happens during some period of time. Moreover, future expansions of the actual facility, is a key issue, and it must be considered.
- In my opinion, several aspects about the modelling process should be included: How is considered the relationship between the size of the facility and the energy losses? How long do you simulate? How do the first point, or the initialization conditions affect or could affect the results? What program do you use? I consider essential to clarify these aspects associated with the algorithm.
- You should detail the implementation of the algorithm, I consider that it is basic to be able to reproduce and validate your results.
- The conclusions are correct, but in my opinion lines 345-350 are banal, this sentences make up a short summary, but not conclusions. You should go deeper into the most important aspects, and analyse more in deep for obtain more rigorous and useful research results.
Reviewer 2 Report
The work is very interesting and concerns extremely interesting issues related to the production of electricity, heat and energy storage. The subject matter is important and very up to date. Nevertheless, the proposed method of using linear programming seeking optimal values of storage parameters is based on one and only one case. In my opinion, more cases should be considered to create a more general method because the considered area is characterized by specific energy demand and individual specificity of energy production. However, I am aware of the difficulty of obtaining a larger group of objects that should be analyzed. That is why I recommend the reviewed article for publication in the journal "Applied sciences"
